# The Influence of Aerobic Type Exercise on Active Crohn’s Disease Patients: The Incidence of an Elite Athlete

**DOI:** 10.3390/healthcare10040713

**Published:** 2022-04-12

**Authors:** Konstantinos Papadimitriou

**Affiliations:** Physical Education and Sport Science, Aristotle University of Thessaloniki, 54636 Thessaloniki, Greece; kostakispapadim@gmail.com; Tel.: +30-69-8026-5800

**Keywords:** inflammation, bowel, malnutrition, training, physical activity

## Abstract

A lifestyle factor which contributes to the remission of Crohn’s disease (CD) is physical activity. The effect seems to positively impact the disease’s symptoms, improving the quality of life, especially on patients in remission. Due to the lack of clinical studies about the effects of physical activity on active CD patients, the purpose of the present case study was to record the influence of swimming training (aerobic type of exercise) on an athlete with active CD. In this study participated a 22-year-old male, who is an elite swimmer and who was diagnosed in 2019 with CD. The research was conducted over the last three years (2019–2022). Both the athlete and doctor consented to the clinical examinations by the author. According to the present study, immediate medical examination and the prescription of anti-TNF-α therapy is probably the most appropriate solution for someone who is diagnosed with CD symptoms. Moreover, patient participation in any sport activity is discouraged because of the potential danger of exacerbation of the symptoms. Therefore, for the sake of patient safety, physical activity should only be encouraged when the disease is in remission.

## 1. Introduction

Inflammatory Bowel Disease (IBD) is an autoimmune disease which mainly affects the gastrointestinal tract, especially in young adults though rarely on children [1]. IBD is categorized into ulcerative colitis (UC) and Crohn’s disease (CD). In CD, the symptoms are diarrhea, abdominal pain, urgency, fatigue, weakness, anorexia, and malnutrition, altering body composition [2,3,4]. A lifestyle factor that possibly contributes to the remission of the disease is physical activity.

The research was conducted by the author supported the hypothesis that physical activity contributes to the disease’s remission [5]. More specifically, moderate to intense aerobic and/or resistance exercise, (60–80%, of VO2max or 1 RM), with an interval of 15–30 s and 2–3 min after each exercise, and between exercises respectively, reduces CD’s symptoms, while improving the patient’s quality of life, especially on the disease’s remission [5,6,7]. However, a lack of clinical studies about the effect of physical activity on active CD patients is observed. As a result, doctors and physical activity instructors avoid the prescription of any type of exercise on active CD patients because of the belief that it could lead to the exacerbation of the disease [5,8,9,10,11,12].

Therefore, the purpose of the present case study was to monitor the influence of swimming (aerobic type of exercise) on an active CD’s patient via the incidence of an adult male elite athlete who was diagnosed with CD. The research was conducted over the last three years (2019–2022) from when he was diagnosed with CD. Both athlete and doctor consented to the clinical exams carried out by the author.

## 2. Case Report

The present study is concerned with an elite 22-year-old male swimmer who was diagnosed with CD in 2019. This affected his performance and the will to participate at the training sessions in a sport which he begun in 2005 and continues still. To start with, in 2018, at his senior year as a high school student and during a tough training period, he manifested symptoms of diarrhea which ceased after three days of recovery, improving his condition. Thus, the doctor suggested that probably it was gastroenteritis. Since the appearance of these symptoms, he followed his schedule and training with under normal conditions.

In January 2019, after his dinner, which was a slice of pizza, he felt intense abdominal pain and a few hours later, manifested symptoms of diarrhea. Over the following two weeks since the first manifestation of diarrhea, his dietary and training schedule were performed normally. However, the symptoms of diarrhea continued, disturbing his daily routine. According to his clinical situation, the doctor concluded that the diarrhea was probably due to gastroenteritis; therefore, a dietary schedule was recommended that avoided mainly high-fat foods, such as processed meat, pork products, and fast food, etc., which could have exacerbated the diarrhea. Moreover, he felt more confident to consume mainly carbohydrate foods and poultry, while avoiding any kind of leafy vegetables, fruits, fishes, or legumes, etc. Despite the modifications to his diet, the manifested symptoms (abdominal pain and diarrhea) remained, and he continued training, although his body composition was unaffected.

In the February of the same year, his clinical situation had not improved, and he suffered from strong abdominal pain which affected his bodyweight, resulting in the loss of 4 kg. The doctor recommended a hematological analysis to examine the value of C—Reactive Protein (CRP), a biomarker which is used for the detection of possible inflammation [13]. According to the analysis, the athlete had mild inflammation, with the CRP value at 6.2 mg/dL. Thus, the doctor hypothesized that this inflammatory condition originated from the intestines. Therefore, an antibiotic treatment was administrated, with three tabs per day of Metronidazole.

In March, a month later, the athlete did not show any improvement since the beginning of the treatment. His body weight decreased more than 12 kg since January (body weight 69 kg) and the CRP was raised to 13.87 mg/dL. As a result, it was difficult for him to continue training at the same frequency and intensity as before despite his intentions to do so. According to the literature [5], this was due to the exacerbation of abdominal pain and diarrhea. Thus, in April, he completely stopped training and any other physical activity due to the weakness that he was feeling.

In the same month (April), he did an endoscopy of the colon. The results showed mild inflammation and mucosal atrophy. Therefore, the doctors observing his clinical situation, and in accordance with the histological examination, prescribed him a new treatment combining Metronidazole with the anti-inflammation Mesalazine (two tabs per day).

The two medications (Metronidazole and Mesalazine), a month later (May), inhibited the value of CRP to 0.14 mg/dL, improving his health significantly. As a result, he was able to resume the training, though at a relatively low frequency and intensity. Although his clinical situation improved, his performance was still lower than before, and the exacerbations continued. In the middle of the month, Mesalazine’s treatment was discontinued and replaced with the antibiotic Ciprofloxacin (6 tabs per week) which was administrated until the end of May.

In June, the athlete’s body weight and CRP value returned to almost normal levels (75 kg and 0.38 mg/dL, respectively). However, because of some exacerbations (abdominal pain and diarrhea) the doctor decided to examine the fecal calprotectin (CAL), which is a highly effective index to detect possible endoscopic ulcerations in CD [14]. Furthermore, a colonoscopy was performed and biopsies were taken from the colon.

According to the analysis, the excreta’s inflammation index was 780 μg/g. Likewise, from the colonoscopy examination, the sigmoid and ileum colon showed very red parts with the presence of exudate ulcers. Finally, the results from the biopsies showed long-term active ileitis. Connecting the results of colonoscopy (Figure 1A,B), CAL, and the clinical situation of the athlete, the doctor’s hypothesis of a mild CD infection was confirmed. Despite the athlete’s situation, the doctor recommended that he observe his nutrition intake and to continue with Salofalk at a dose of 500 mg (4 tabs per day). Moreover, his training continued at a lower frequency and intensity, although it remained difficult to sustain the training for more than 30 min (Figure 2).

The athlete, after four months (July–October) of an unstable health condition, was, in November of 2019, hospitalized. After a sequence of hematological analysis and in accordance with his medical history, it was decided to begin the injection of anti-TNF-α (biological therapy) with Infliximab and Azathoprine. The first three therapies were conducted in a period of three months (from November of 2019 until January of 2020). From the March of 2020, until the present time, his therapy continues to take place every two months. In November of 2020, another colonoscopy was performed in which a physiological depiction of the colon instead of a mild inflammation in the sigmoid colon was found, which did not concern the doctors. Since the athlete started the therapy, his CRP value reached normal levels (<0.50 mg/dL) (Figure 3). The only exception was in August of 2021, when the CRP was raised above normal values. However, the athlete did not manifest any symptoms of abdominal pain or diarrhea. The doctor indicated that this raise was probably due to another factor, because CRP is an index which is associated with a variety of etiologies, ranging from sleep disturbances to periodontal disease [15]. Finally, his swimming performance was improved significantly when competing again in races.

## 3. Discussion

This novel case study was performed with no limitations or difficulties. Both the athlete and the doctors contributed, giving reports about the hematological analysis and the clinical situation during the three-year period.

According to the athlete’s situation, it is assumed that in the condition of an active disease, aerobic types of exercise exacerbate the disease; however, in remission, aerobic types of exercise do not cause exacerbation. Thus, the results of the present case are in accordance with the literature [5,6,7]. Specifically, many authors regard that aerobic or resistance type of exercise effectively mitigate CD’s symptoms, especially when the patient is in remission. However, there is not any observation in the research of a case in which someone is in active disease’s condition. Thus, there is only the hypothesis that any kind of exercise must be avoided when the symptoms are active [5].

Moreover, the immediate performance of a hematological analysis and colonoscopy after any suspicious CD symptom seems to be important for any patient. Specifically, since the first manifestation of the disease, the doctor administrated to the patient Metronidazole and Mesalazine two months later. For the athlete, these two months were a period which caused an extensive malnutrition and resulted in difficulty in the participation of his training or indeed of any other activity. Therefore, to deal with those types of symptoms, immediate examination, and offensive therapy with anti-TNF-α injection (Infliximab and Azathoprine) are the most appropriate actions [16]. In case of the athlete of this study, the doctors prescribed him medications at the beginning of symptoms’ manifestation as a unique therapy. However, the most effective and appropriate way to deal with CD symptoms is the combined prescription of medication and anti-TNF-α therapy.

The present study is the first which recorded the incidence of an active CD athlete who participates in aerobic activity. Therefore, further studies are essential to inform doctors and physical activity instructors about the potential benefits of the prescription of exercise on CD patients.

## 4. Conclusions

According to the present study, immediate medical examination and prescription of anti-TNF-α therapy are probably the most appropriate solution for someone who is diagnosed with CD symptoms. Moreover, patient participation in any sport activity is discouraged because of the potential danger of exacerbation of the symptoms. Therefore, for the sake of patient safety, physical activity should only be encouraged when the disease is in remission.

## Figures and Tables

**Figure 1 healthcare-10-00713-f001:**
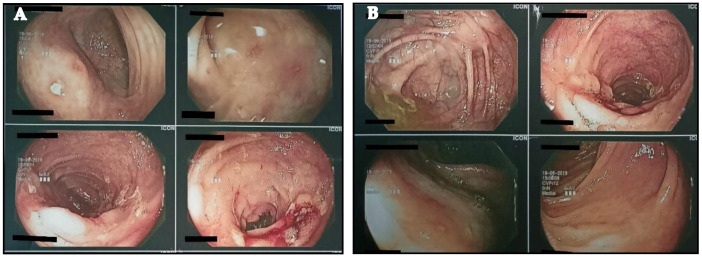
(**A**,**B**) Depiction of colon via colonoscopy.

**Figure 2 healthcare-10-00713-f002:**
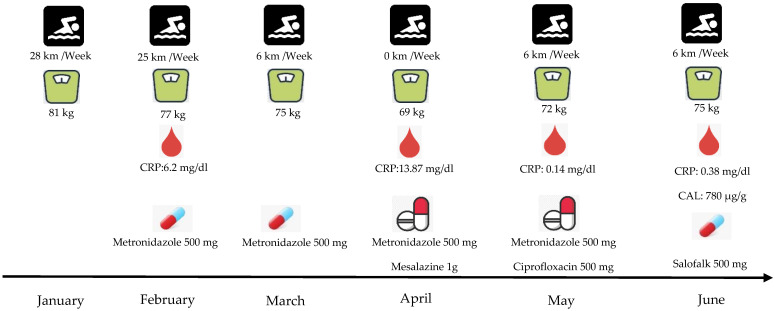
Patient’s clinical situation during the first six months of CD manifestation. CRP: C—Reactive Protein, CAL: Fecal Calprotectin.

**Figure 3 healthcare-10-00713-f003:**
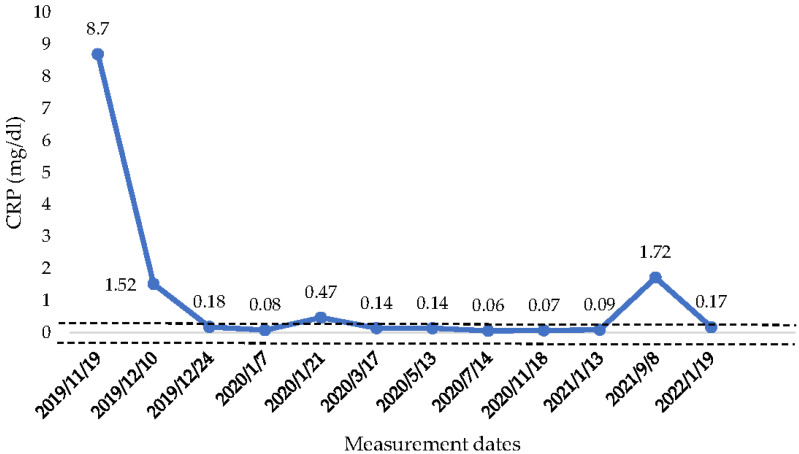
CRP concentration from the beginning of anti-TNF—α therapy until January 2022. CRP: C—reactive protein, 
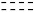
: Range of normal values 0–0.50 mg/dL.

## Data Availability

Not applicable.

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
