# Peer review of "The Influence of Aerobic Type Exercise on Active Crohn’s Disease Patients: The Incidence of an Elite Athlete"

_healthcare, 2022, doi:10.3390/healthcare10040713_

Round 1

Reviewer 1 Report

Review Report

Comment-1- It is not clear from when to when the athlete wanted to participate in the training sessions.

“In the present study, it is demonstrated the incidence of a 22–years old, male, elite swimmer who was diagnosed, in 2019, with CD affecting his performance and the will to participate at the training sessions 12 years since his first participation in the sport”

Comment-2 –The context is not clear. Did the athlete’s condition improve after 3 days?

“To start with, in 2018 at his senior year as a high school student and during a tough training period, he manifested diarrhea symptom which was collapsed after three days of recovery thus”

Comment-3 The author mentioned that the athlete was recommended to avoid fat-free diet. It is important to know whether avoiding the diet improved the diarrhea condition. Also, include the dietary regimen, so that it can help other CD patients.

“Therefore, it was recommended a dietary schedule avoiding, mainly, high fat foods which could probably exacerbate diarrhea. Independently the manifested symptoms, he continued swimming training, whereas his body composition was unaffected.”

Comment-4-Provide an explanation of why does CRP level went up to 1.72 mg/dL on 9/8/2021?

Comment-5- Please fix the legend number

Comment-6- This is not clear.

“Also, medication must be using combined with anti-TNF – α injection and no as a unique therapy.”

Comment-7-It will be worthwhile to indicate athlete's dietary regimen throughout the study time (if available)

Author Response

I really appreciate your helpful comments which were contributed to improve my study. I submit the answers under the comments (C – A). 

C.1. It is not clear from when to when the athlete wanted to participate in the training sessions.

A.1. I specify the period during which the athlete started swimming.

“In the present study, it is demonstrated the incidence of a 22–years old, male, elite swimmer who was diagnosed, in 2019, with CD affecting his performance and the will to participate at the training sessions in a sport which he begun in 2005 and still continues.”

C.2. –The context is not clear. Did the athlete’s condition improve after 3 days?

A.2. Yes, his health was improved.

“To start with, in 2018 at his senior year as a high school student and during a tough training period, he manifested diarrhea symptom which was collapsed after three days of recovery, improving simultaneously his health condition.”

C.3. The author mentioned that the athlete was recommended to avoid fat-free diet. It is important to know whether avoiding the diet improved the diarrhea condition. Also, include the dietary regimen, so that it can help other CD patients.

A.3. There was not any specific dietary program which he followed. However, I give some extra information about it and his diarrhea symptoms.

“Therefore, it was recommended a dietary schedule avoiding, mainly, high-fat foods such as processed meat, porky products, fast food, etc., which could probably exacerbate diarrhea. Independently the manifested symptoms (abdominal pain and diarrhea), which still were active, despite his dietary modification, he continued swimming training, whereas his body composition was unaffected.”

Comment-4-Provide an explanation of why does CRP level went up to 1.72 mg/dL on 9/8/2021?

A.4. I give the explanation in the text.

“The only exception was in August of 2021 in which CRP was raised above the normal values. However, the athlete did not manifest any symptoms of abdominal pain or diarrhea. Also, the doctor indicated that probably this raise is due to another factor because CRP is an index which is associated with a variety of etiologies, ranging from sleep disturbances to periodontal disease [16].”

C.5. Please fix the legend number

A.5. Done

Comment-6- This is not clear.

A.6. Now I think is clearer.

“In athlete’s incidence, the doctors prescribed him, at the beginning of symptoms’ manifestation, medications as a unique therapy. However, the most effective and appropriate way to deal with CD symptoms is the combined prescription of medication and anti-TNF – α therapy.”

C.7. It will be worthwhile to indicate athlete's dietary regimen throughout the study time (if available).

A.7. I understand the meaning of that information. However, the athlete did not follow any planned dietary schedule. The only thing that I can add as information is his most preferable foods (second paragraph of “Case study” section”.

Also, he was feeling more secure to consume mainly carbohydrate foods and poultry, avoiding any kind of leaf vegetables, fruits, fishes, legumes, etc.”

Reviewer 2 Report

Dear author,

It is a well written manuscript, it need only few modiffication.

Please, check the scienetific soundness or the reference of this first sentenece, second paragraph from case report:  On January .....

 I recommend authors cite the following paper as it contains very interesting information : Low doses of mesalamin taken on regularly base have prophylactic effects in diverticular disease WOS:000295099200331

Section Discussion may be improved, i think it s short

Author Response

I really appreciate your helpful comments which were contributed to improve my study. I submit the answers under the comments (C – A). 

C.1. Please, check the scienetific soundness or the reference of this first sentenece, second paragraph from case report:  On January .....

 A.1. Checked. In the sentence is described the aitiology of disease’s activation.

C.2. I recommend authors cite the following paper as it contains very interesting information : Low doses of mesalamin taken on regularly base have prophylactic effects in diverticular disease WOS:000295099200331

A.2. The reference does not exist in the form that you write it.

C.3. Section Discussion may be improved, i think it s short

A.3. Was added an extra paragraph keeping the key points of the discussion.

Round 2

Reviewer 1 Report

The author addressed all the concerns, therefore it can be accepted in the current format.

Author Response

Thank you for your reply.